# Versatile non-luminescent color palette based on guest exchange dynamics in paramagnetic cavitands

Elad Goren[1], Liat Avram[2] & Amnon Bar-Shir [1✉]

Multicolor luminescent portrayal of complexed arrays is indispensable for many aspects of science and technology. Nevertheless, challenges such as inaccessible readouts from opaque objects, a limited visible-light spectrum and restricted spectral resolution call for alternative approaches for multicolor representation. Here, we present a strategy for spatial COlor Display by Exploiting Host-guest Dynamics (CODE-HD), comprising a paramagnetic cavitand library and various guests. First, a set of lanthanide-cradled α-cyclodextrins (**Ln-CDs**) is designed to induce pseudo-contact shifts in the $^{19}$F-NMR spectrum of **Ln-CD**-bound guest. Then, capitalizing on reversible host-guest binding dynamics and using magnetization-transfer $^{19}$F-MRI, pseudo-colored maps of complexed arrays are acquired and applied in molecular-steganography scenarios, showing CODE-HD's ability to generate versatile outputs for information encoding. By exploiting the widely shifted resonances induced by **Ln-CDs**, the guest versatility and supramolecular systems' reversibility, CODE-HD provides a switchable, polychromatic palette, as an advanced strategy for light-free, multicolor-mapping.

---

[1] Department of Molecular Chemistry and Materials Science, Faculty of Chemistry, Weizmann Institute of Science, Rehovot, Israel. [2] Department of Chemical Research Support, Faculty of Chemistry, Weizmann Institute of Science, Rehovot, Israel. ✉email: amnon.barshir@weizmann.ac.il

The palette of spectrally resolved luminescent colors provides us the ability to distinguish between dyed objects, and therefore, to observe and study complex milieus, where multiple components are indistinguishable in their "monochromatic" appearance. Thus, numerous color signatures and multiplexed readout features stimulated the development of advanced materials, used not only to tackle challenges in basic sciences, but also for advanced applications[1–4]. Indeed, illuminating (small) molecules[5], polymers[6], nanoparticles[7], and proteins[8] are all channels for the design of a broad palette of orthogonal and spectrally resolved colored materials, which are employed widely —from a better understanding of complex systems[9] to sensing[10] and logic gates[11], through molecular steganography[12], imaging of live organisms[13], and up to electronic devices[14]. Nevertheless, several challenges remain for luminescent materials, including photobleaching, light penetration through nontransparent media, a limited number of spectrally resolved colors, color palette transformability (e.g., extendibility to additional colors and their deletion), through-object tomography-based mapping capabilities, and more. Therefore, nonluminescent strategies that can capture the polychromatic capabilities of light could expand the material-based designs for multiplexing detection and multispectral display in scenarios beyond those that are applicable for luminescence-based colors.

The introduction of supramolecular chemistry[15–17] has opened numerous possibilities for designing noncovalent dynamic assemblies, resulting in the establishment of synthetic supramolecular systems as accessible tools for both fundamental research and advanced applications. Among the developed supramolecular architectures, those that are composed of large hosts and small molecular guests were developed and used in many disciplines[18–24]. Taking advantage of the dynamic nature of host–guest systems and exploiting the chemical shift difference in NMR spectra, a unique method for magnetization transfer between a host-bound and free guest molecules was proposed[25]. This method, based on chemical exchange saturation transfer (CEST), which is frequently used for MRI mapping of solutes through a proton exchange process[26], was shown to be applicable for host–guest systems that are composed of Xe gas and a variety of large hosts through an advanced hyperpolarized $^{129}$Xe-NMR setup (i.e., hyperCEST)[27–32]. The extension of CEST into $^{19}$F-NMR[33], together with the diversity of potential fluorinated guests, to obtain the host–guest variant of CEST, termed GEST[34–36] (guest exchange saturation transfer), created the

opportunity to design large number of innovative supramolecular platforms.

Depending on the different chemical shift offsets of exchangeable protons of different solutes and their fast exchange with water protons, artificial colors were produced in $^{1}$H-CEST-MRI maps for biomedical applications[37,38]. The ability to induce pseudo-contact shifts (PCSs) to exchangeable protons and introducing the paramagnetic CEST (paraCEST)[39,40] approach, has further expanded the spectrally resolved pseudo-colors for MRI applications[41]. The multicolor features shown by hyperCEST[27] and paraCEST MRI[39,40], together with the ability to implement the PCS principles to $^{19}$F-NMR[42] and $^{19}$F-MRI[43] inspired us to develop a supramolecular palette for COlor Display by Exploiting Host–guest Dynamics (CODE-HD).

Here, we show the design, development, characterization, and implementation of a supramolecular system that generates multispectral pseudo-colors based on host–guest exchange kinetics, paramagnetic-induced PCS, and CEST contrast in a $^{19}$F-MRI framework. This method enables multiplexed information encoding and multicolor displays of supramolecular systems without the need for a light source. As a proof-of-concept, we show the potential use of CODE-HD for molecular steganography applications, demonstrating its high level of security, versatility (switchable color palette), and eraseability. Finally, based on the principles of tomography-based techniques, MRI among them, CODE-HD is utilized to simultaneously encode different layers of information from a single studied object.

## Results

**Paramagnetic cavitands design.** Introducing the paramagnetic GEST (paraGEST) approach, in which the obtained chemical shift ($\Delta\omega$) of the bound guest depends on the paramagnetic element in the host (Fig. 1a), a library of paramagnetic cavitands (para-CDs) was first constructed. To this, 6$^A$,6$^D$-diamino-6$^A$,6$^D$-dideoxy-α-cyclodextrin (**diamino-CD**, Supplementary Figs. 1 and 2) was conjugated to a lanthanide chelating cradle, diethylenetriaminepentaacetic dianhydride (DTPAA), to obtain **CD-DTPA** (α-CD-diethylenetriaminepentaacetic acid, Fig. 1b), through their reaction in anhydrous dimethyl sulfoxide (16 h stirring) in the presence of triethylamine. The resultant white powder was collected after centrifugation and purified using preparative reversed-phase high-pressure liquid chromatography (HPLC). The purity of the obtained **CD-DTPA** product was evaluated by

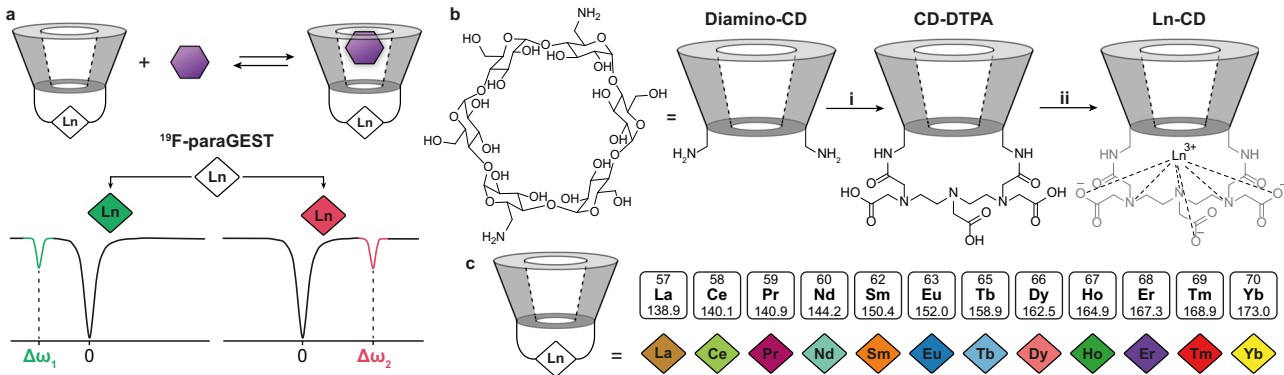

**Fig. 1 COlor Display by Exploiting Host–guest Dynamics (CODE-HD) rationale and lanthanide-cyclodextrins (Ln-CDs) library fabrication. a** The dynamic exchange process that allows a paramagnetic guest exchange saturation transfer (paraGEST) observation and two schematic z-spectra showing different chemical offsets ($\Delta\omega_1$, green, and $\Delta\omega_2$, pink) values, when using different Ln-CDs (green and pink diamonds). **b** The synthetic route used for the synthesis of Ln-CD: (i) diethylenetriaminepentaacetic dianhydride and triethylamine, in anhydrous dimethyl sulfoxide (DMSO) at room temperature for 16 h; (ii) reflux in aqueous Lanthanide chloride (LnCl$_3$) solution for 1 h, -hydrochloric acid (HCl). **c** The constructed Ln-CD library with the different Ln used represented as color diamonds.

an analytical HPLC (Supplementary Fig. 3), followed by its full characterization using a set of one-dimensional (1D; $^1$H-, $^{13}$C-) and two-dimensional (2D; $^1$H–$^1$H COSY, $^1$H–$^{13}$C HSQC, and $^1$H–$^1$H ROESY) NMR experiments (Supplementary Discussion, Supplementary Figs. 4–19, and Supplementary Tables 1 and 2) and high-resolution mass spectrometry (Supplementary Figs. 20 and 21). Then, based on the procedure applied to obtain lanthanide-cradled β-cyclodextrins[44,45], pure **CD-DTPA** was refluxed in water in the presence of different lanthanide chloride salts (LnCl$_3$, composed of different Ln$^{3+}$ cations) to obtain the desired set of lanthanide-cyclodextrins (**Ln-CDs**, Fig. 1c and Supplementary Figs. 22–40) as CODE-HD's hosts. In addition to the paramagnetic **Ln-CDs** that are expected to induce variable PCS effects upon guest complexation, **La-CD** was synthesized as a diamagnetic host for complementary NMR experiments. Note here that the obtained aminopolycarboxylate **CD-DTPA** should have a strong binding affinity toward lanthanides in the resultant **Ln-CDs**, as those obtained for the clinically used Gd-DTPA-BMA contrast agent[46]. Nevertheless, as the binding affinities may slightly deviate for different Ln$^{3+}$, as reported for different Ln-DTPA-BMAs[47,48], further stability studies and safety profiles should be obtained prior to the use of **Ln-CDs** in any biological application in the future.

**ParaGEST characterization of host–guest complexes.** Having established an array of **Ln-CDs**, their potential use as the paramagnetic hosts in CODE-HD was examined with four putative guests (**1**–**4**), which share the same benzyl amine backbone (Fig. 2a). The formation of an inclusion complex between the guest and **Ln-CD**, essential for magnetization transfer, was confirmed by a 2D $^1$H–$^1$H ROESY experiment performed on a solution containing guest **1** and the diamagnetic host **La-CD** (Fig. 2b)[49]. Then, a set of $^{19}$F-GEST NMR experiments was acquired for samples containing the paramagnetic host **Dy-CD** and each of guests **1**–**4**, revealing a clear dependency of the Δω of the obtained paraGEST effect on the structure of the guest. The relatively large chemical shift of the GEST peaks clearly manifests the PCS induction of the paramagnetic Dy on the $^{19}$F-NMR resonance of a bound guest, as indicated by Δω values of −20.6 ppm (Fig. 2c), −18.2 ppm (Fig. 2d), −28.5 ppm (Fig. 2e), and −61.9 ppm (Fig. 2f) for guests **1**, **2**, **3**, and **4**, respectively. Such differences in the Δω values and the magnification ("dips") of the paraGEST effects can be attributed to the distance (and angle) between the affected nucleus (fluorine) and the lanthanide (Dy$^{3+}$)[50], but may also be the result of steric hindrances that mediate exchange rate and guest inclusion. Importantly, an analog of guest **1**, which is substituted with a carboxylic acid instead of a primary amine group, also yielded a pronounced paraGEST effect (Supplementary Fig. 41a), while other analogs, substituted with hydroxyl or amino-methyl functional groups, did not generate any noted effect (Supplementary Fig. 41b–d). These observations reflect the cruciality of a functional group with Ln$^{3+}$-coordination capabilities (primary amine or a carboxylic acid) for a successful magnetization transfer effect (Fig. 2 and Supplementary Fig. 41). Based on the PCS induced by the paramagnetic element at the center of the cavitand, we expected that CODE-HD would provide high spectral resolution with no paraGEST profile overlaps upon the use of different lanthanides. Thus, utilizing an array of several para-CDs (Fig. 1), a single significant paraGEST effect in $^{19}$F-NMR (Fig. 2) could be further extended into a frequency-encoded palette that has the potential to be displayed in a multicolor manner (i.e., CODE-HD).

**CODE-HD construction.** Relying on the three equivalent fluorine atoms of guest **2** that should result in a better signal-to-noise

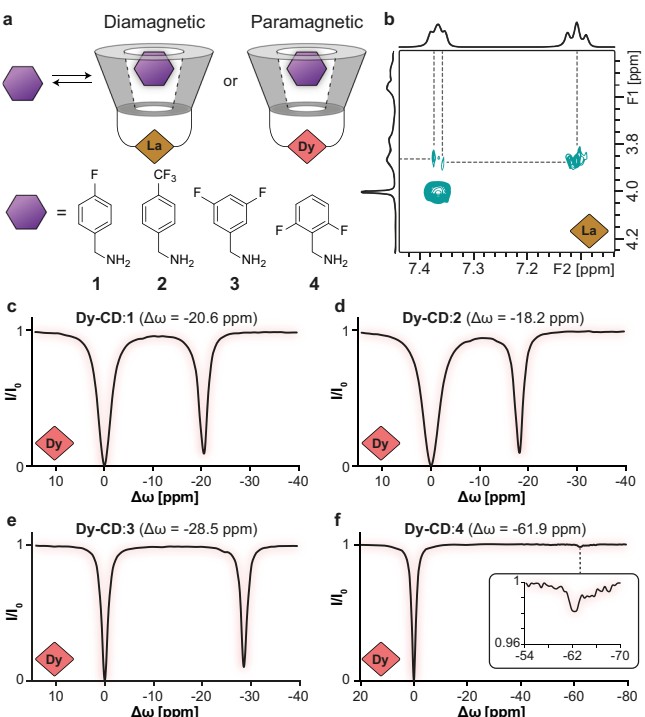

**Fig. 2 Paramagnetic guest exchange saturation transfer (paraGEST) nuclear magnetic resonance (NMR) of variable guests. a** Schematic illustration of the diamagnetic lanthanum-cyclodextrin (**La-CD**, brown diamond) and paramagnetic dysprosium-cyclodextrin (**Dy-CD**, peach diamond) hosts and guests **1**–**4** (represented by a purple hexagon) used to determine guest inclusion and paraGEST characteristics. **b** A section of the proton–proton rotating frame overhauser effect spectroscopy ($^1$H-$^1$H ROESY) spectrum acquired from an aqueous solution of **1** and **La-CD**. The z-spectra obtained from aqueous solutions containing the paramagnetic host **Dy-CD** and guests **2** (**c**), **2** (**d**), **3** (**e**), and **4** (**f**, inset shows magnification of the paraGEST effect at −61.9 ppm).

ratio and given its large GEST effect, **2** was used as the putative guest for the initial examination of CODE-HD performances. To this end, nine CODE-HD pairs composed of guest **2** and different para-CDs (Fig. 1c) were examined for their $^{19}$F-paraGEST NMR characteristics. An array of single and well-defined paraGEST effects was obtained for all examined **Ln-CD:2** pairs with characteristic Δω values, which depend on the identity of the Ln$^{3+}$ center of the host (Fig. 3a). Performing the magnetization transfer asymmetry (MTR$_{asym}$) analysis (Supplementary Fig. 42) for each host–guest pair, and further assigning the Δω values of each paraGEST effect with color (Fig. 3b and Supplementary Table 3) resulted in the establishment of the pseudo-color palette of CODE-HD. Note here, as expected for lanthanides with poor PCS capabilities (i.e., **Ce-CD** and **Sm-CD**), relatively small Δω values were obtained in the paraGEST spectra (Supplementary Fig. 43f and Supplementary Fig. 43g, respectively), and therefore, these paramagnetic hosts were not included in the proposed CODE-HD palette. Nevertheless, these hosts may offer further supramolecular colors in the future, when paired with a different guest molecule or when using MRI scanners operating at stronger magnetic fields.

To demonstrate CODE-HD in a 2D manner, each well of a 3 × 3 multiwell plate was loaded with an aqueous solution of guest **2**, in the presence of a different paramagnetic host (paraGEST library, Fig. 3c and Supplementary Table 4), and the plate was placed in a 15.2 T MRI scanner. Both $^1$H-MRI (Fig. 3d) and $^{19}$F-MRI (Fig. 3e) images of the scanned plate resulted in no significant difference

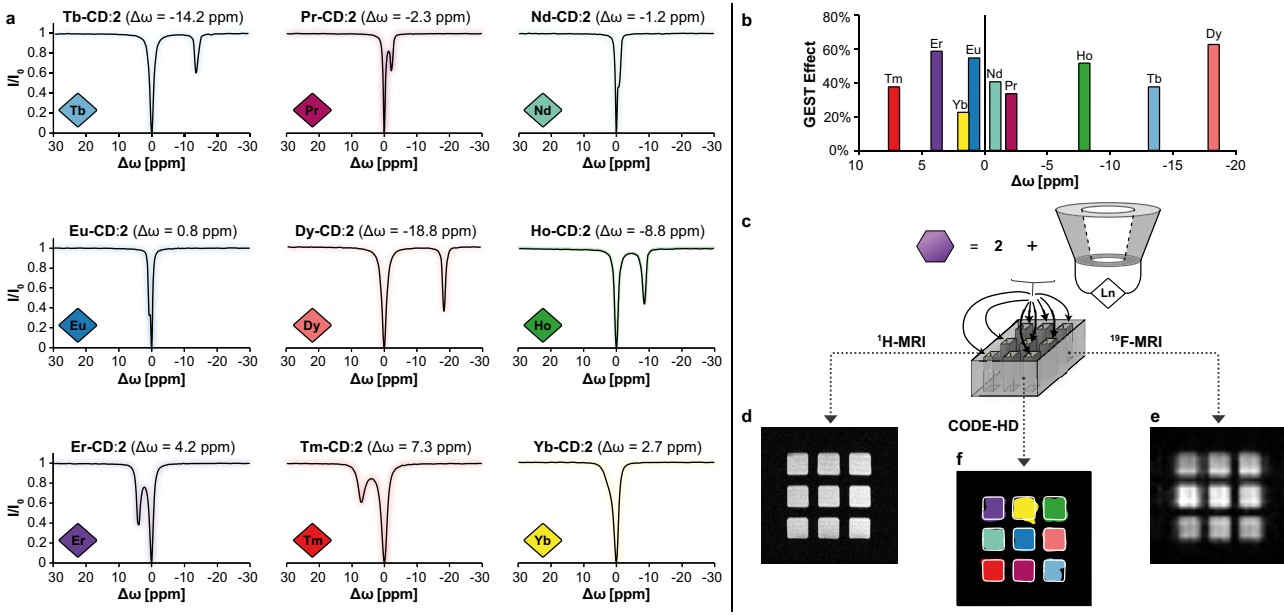

**Fig. 3 COlor Display by Exploiting Host–guest Dynamics (CODE-HD) construction and pseudo-color assignment. a** z-Spectra of aqueous solutions containing lanthanide-cyclodextrin (**Ln-CD**) hosts (represented as color diamonds) and guest **2**. **b** Color-coded guest exchange saturation transfer (GEST) effect plots of the studied library shown in **a**. **c** Schematic illustration of the obtained CODE-HD setup of a $3 \times 3$ multiwell plate loaded with nine different **Ln-CD** hosts (one host/well) and guest **2**. Shown are proton magnetic resonance imaging ($^1$H-MRI) (**d**), $^{19}$Fluorine-MRI ($^{19}$F-MRI) (**e**), and CODE-HD map (**f**), where each color represents the chemical shift offset ($\Delta\omega$) of each **Ln-CD**:**2** paramagnetic GEST effect (paraGEST, the color-code shown in **b** and assigned in Supplementary Table 3).

between the MRI appearance of the nine wells. However, acquiring $^{19}$F-GEST MRI data and performing an MTR$_{asym}$ analysis (Supplementary Fig. 44) resulted in a colored CODE-HD map (Fig. 3f). In this manner, the location of each **Ln-CD**:**2** pair can be spatially depicted and assigned a pseudo-color, according to the $\Delta\omega$ of the observed paraGEST effect (Fig. 3b, f and Supplementary Table 3). Thus, applying CODE-HD in an MRI setup, we demonstrated the ability to spatially map nine spectrally resolved pseudo-colors through a 2D color display of their location. These nine "markers" can be used in a variety of potential combinations that could be arranged in numerous displayable patterns for diverse applications in several fields, beyond biomedicine.

**Molecular steganography applications.** One potential application of CODE-HD is demonstrated through molecular steganography, where combinations of hideable, modifiable, and erasable[51–54] spectrally resolved colors may be used to secure encoded information. For example, in a simple case where a single host–guest pair is placed in each one of the given nine wells (Supplementary Table 5), CODE-HD can be used to generate color-based barcodes or patterns (Fig. 4a, b), in the same fashion shown for fluorescent-based molecular systems[53,55]. For this particular use, the order in which the $^{19}$F-paraGEST data are acquired is not essential for revealing the hidden pattern, as each well contains only a single host. Note here that to obtain these color patterns, the paraGEST data can be acquired for all **Ln-CDs** (all $\Delta\omega$'s) from which only the one included in the plate will generate the related pseudo-color. Given nine artificial colors and nine wells, CODE-HD can be used to encode nine factorial ($9! = 362,880$) different colored barcodes, like those shown in Fig. 4a, b. Notice that future applications, including **Ce-CD** and **Sm-CD**, which induced relatively small $\Delta\omega$ values in **2** (Supplementary Fig. 43), can add additional colors to CODE-HD, enabling even more color combinations (although limited for strong-field MRI scanners, as the one used here, Supplementary Fig. 45). By this,

CODE-HD is able to offer eleven factorial ($11! = 39,916,800$) different combinations in dedicated setups.

In addition and notably, the high spectral resolution of the "colors" of CODE-HD, which is the result of the large PCS induction by the paramagnetic elements of the **Ln-CD** cavitands, can be exploited for more complex encoding strategies, in which multiple host–guest pairs are located in a single well of the $3 \times 3$ plate. This mode of information encoding allows, in principle, 387,420,489 combinations of colors ($9^9$ when arranged in a $3 \times 3$ multiwell plate). To demonstrate this, we capitalized on the ability to generate a $3 \times 3$ typeface of all Latin alphabet letters, 0–9 digits and additional symbols (Supplementary Fig. 46) to encrypt either words (Fig. 4c, d) or PIN codes (Fig. 4e, f). For that purpose, multiple hosts should be loaded into each one of the wells to allow the encoding and decoding of an entire word or PIN code from a single $3 \times 3$ plate.

To this end, each well of the $3 \times 3$ multiwell plate was first loaded with several **Ln-CD** hosts in such a way that a single plate reflected either the hidden word or PIN code (Supplementary Tables 6 and 7). Followed by the addition of guest **2** to all wells and the acquisition of $^{19}$F-paraGEST data in a given order (i.e., $\Delta\omega_1 \rightarrow \Delta\omega_2 \rightarrow \Delta\omega_3 \rightarrow \dots$), the encrypted information was decrypted. Figure 4c depicts an example of the hidden word "CEST"—while the identity of the loaded **Ln-CD** hosts in each well cannot be revealed by either $^1$H-MRI or $^{19}$F-MRI, applying CODE-HD at the $\Delta\omega$ values of $-18.8$ ppm (**Dy-CD**:**2**), $+7.3$ ppm (**Tm-CD**:**2**), $-8.8$ ppm (**Ho-CD**:**2**), and $-2.3$ ppm (**Pr-CD**:**2**), i.e., the code "Dy–Tm–Ho–Pr," revealed the hidden word (CEST). In the same manner, the word "color" was encoded by "Tb–Yb–Er–Yb–Ho", as shown in Fig. 4d. Figure 4e, f shows two examples of secured PIN codes, i.e., "1934" and "*6075" uncovered by the codes "Dy–Ho–Tm–Pr" and "Tb–Pr–Tm–Er–Yb," respectively. Importantly, in contrast to the encrypted colored barcodes (Fig. 4a, b), decrypting hidden words or PIN codes (Fig. 4c–f) without a specific given order (e.g., "Dy–Tm–Ho–Pr" in Fig. 4c and "Tb–Pr–Tm–Er–Yb" in Fig. 4f)

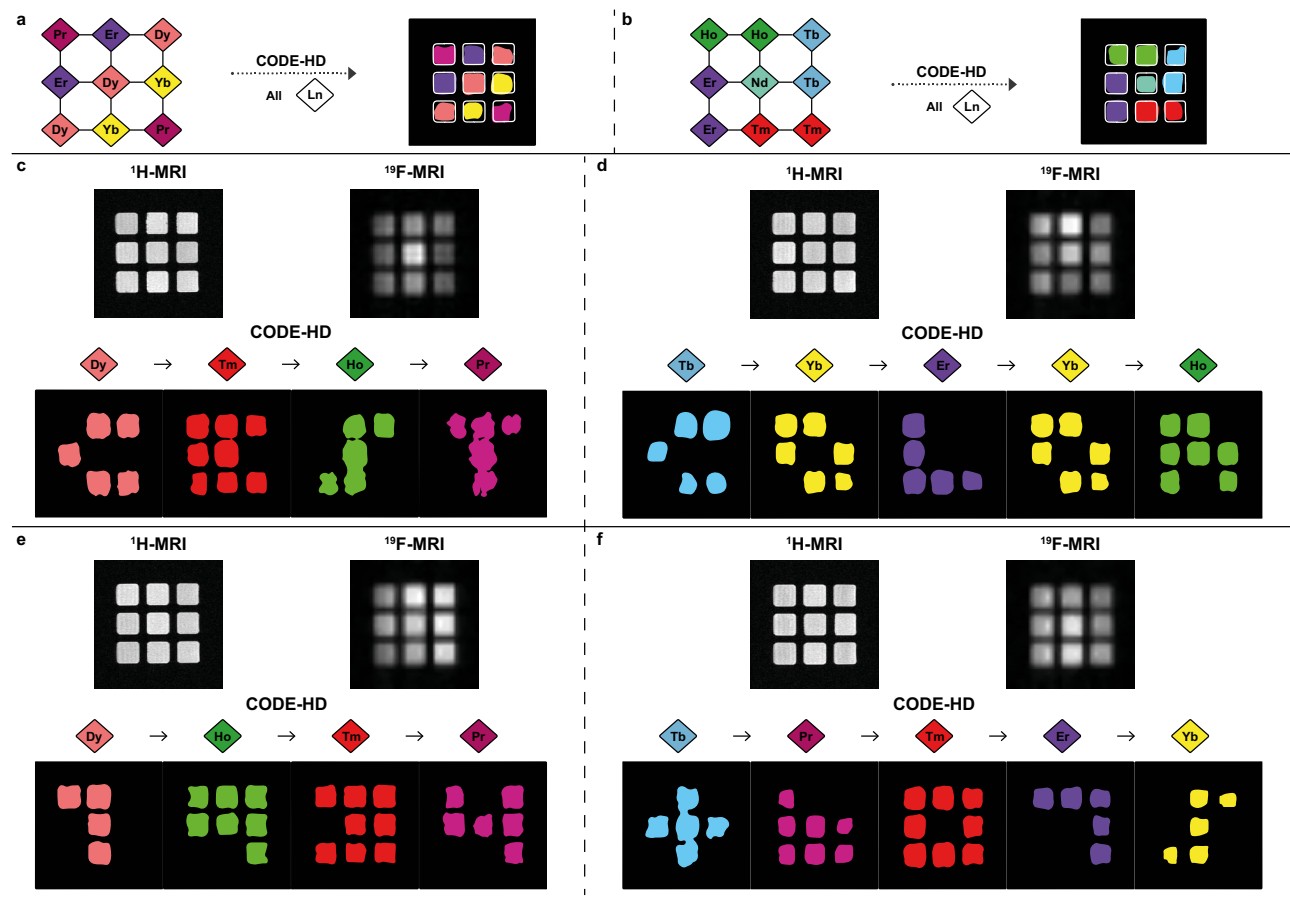

**Fig. 4 Molecular steganography using COlor Display by Exploiting Host–guest Dynamics (CODE-HD) with pairs of lanthanides-cyclodextrins (Ln-CDs):2.** Hidden colored barcodes (**a**, **b**). A given pattern in which a single **Ln-CD:2** pair was loaded in each well of the studied 3 × 3 multiwell plates (diamonds) and the resultant CODE-HD maps (squares), revealing the hidden color-code barcodes. Hidden words (**c**, **d**). Proton magnetic resonance imaging ($^1$H-MRI) and $^{19}$Fluorine-MRI ($^{19}$F-MRI, shown in grayscale) of the studied 3 × 3 multiwell plates and sequential CODE-HD maps obtained in a given order (Dy–Tm–Ho–Pr and Tb–Yb–Er–Yb–Ho for **c** and **d**, respectively), revealing the hidden words (CEST and COLOR for **c** and **d**, respectively). Hidden PIN codes (**e**, **f**). $^1$H-MRI and $^{19}$F-MRI (shown in grayscale) of the studied 3 × 3 multiwell plates and sequential CODE-HD maps obtained in a given order (Dy–Ho–Tm–Pr and Tb–Pr–Tm–Er–Yb for **e** and **f**, respectively), revealing the hidden PIN codes (1934 and *6075 for **e** and **f**, respectively).

will result in a scrambled, unrevealed message. Thus, this demonstration of using CODE-HD for molecular steganography reflects its versatility for encoding multiple information types (barcodes, words, and PIN codes), based on different combinations of colors and patterns, that could be revealed only by users who are familiar with CODE-HD's elements (**Ln-CD** library, identity of the $^{19}$F-guest and $^{19}$F-paraGEST principles). Compared to other MRI-based multicolor approaches, either CEST-based or $^{19}$F-MRI based, where only a single component of the encoding system (the used agent) can be controlled, in CODE-HD, the higher level of security is reflected by the fact that the hidden code is composed of two components (host and guest).

**Multiplexed encoding capabilities.** Following that, to manifest CODE-HD flexibility and versatility, we demonstrated additional advanced abilities, such as the ability to convert a given color-code to another, code-deletion capabilities, and the ability to simultaneously extract different patterns from a sealed, multi-layered volumetric subject. One major advantage of CODE-HD over other molecular systems used for color-encoding is its switch ability. In principle, by replacing the used guest (**2**) with another one, which generates significant $^{19}$F-paraGEST effects in different $\Delta\omega$ values in the presence of the same members of the **Ln-CDs** library (see Fig. 2), an alternative color palette can be generated for CODE-HD (Fig. 5a, b). Specifically, replacing **2** (Fig. 5a) with

**3** (Fig. 5b) as the putative guest and using the same host, e.g., **Tm-CD**, resulted in significantly shifted $\Delta\omega$ values for the obtained $^{19}$F-paraGEST effects, from $\Delta\omega = 7.3$ ppm. (with guest **2**, Fig. 5a) to $\Delta\omega = 17.0$ ppm (with guest **3**, Fig. 5b). Implementing this principle for all **Ln-CDs** in the library, and using guest **3** rather than guest **2**, an additional CODE-HD color palette was generated (Fig. 5b, Supplementary Fig. 47a, and Supplementary Tables 8 and 9). This change provided CODE-HD with a convertible color-code capability—a property that is not achievable for classical luminescence-based colors or for MRI-based artificial colors. Such a feature, which extends the available "colors" of CODE-HD and doubles the number of possibilities for multiplexed information encoding, reflects, again, the platform's versatility through a simple guest replacement.

Another unique feature of CODE-HD is its eraseability. Adding guest **5** (Fig. 5c), which has a high affinity to α-CD[56] and thus prevents **Ln-CD:3** inclusion (Supplementary Fig. 48), leads to the elimination of all the paraGEST effects of **Ln-CD:3** pairs, resulting in the deletion of the multicolor pattern. The nullified $^{19}$F-paraGEST effect in each well is reflected by the elimination of the characteristic peak at the z-spectrum obtained for each well (shown for **Er-CD:3** in Fig. 5c and for other pairs in Supplementary Fig. 47b). These code manipulation capabilities of CODE-HD (Fig. 5a–c) reflect again its flexibility for enhancing both message complexity (i.e., more permutations) and security

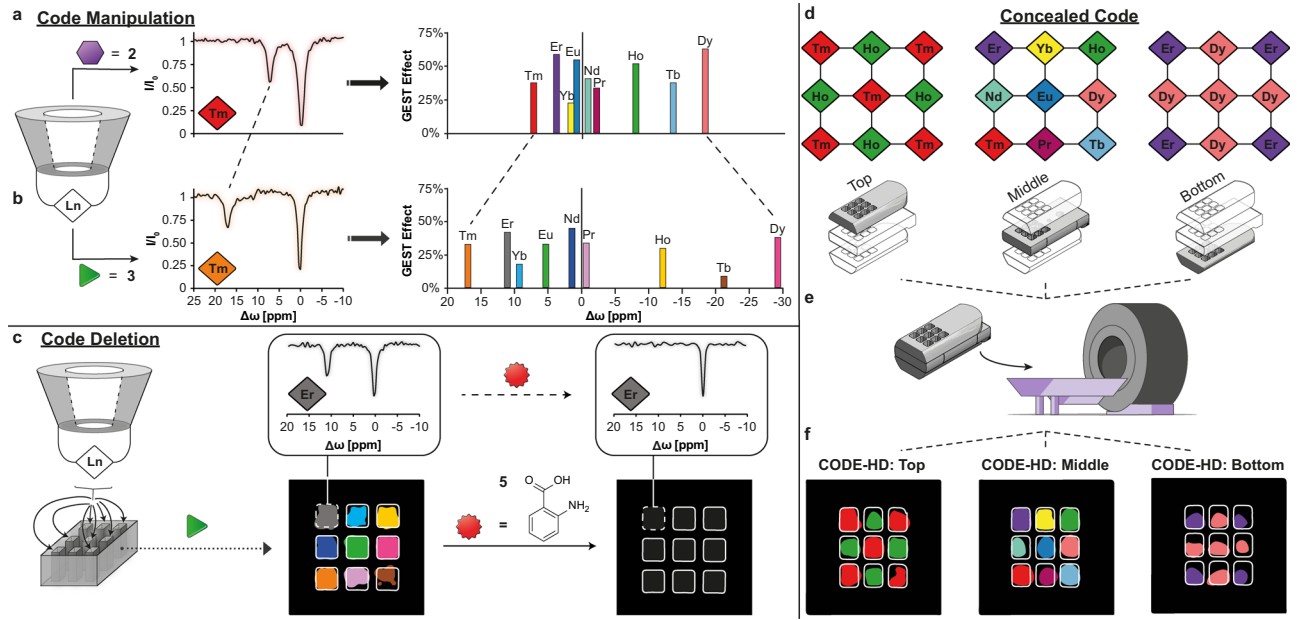

**Fig. 5 Color Display by Exploiting Host–guest Dynamics (CODE-HD) code manipulation.** The obtained z-spectra of a solution of thulium-cyclodextrin (**Tm-CD**, red and orange diamonds) in the presence of either **2** (a, purple hexagon) or **3** (b, green triangle), and plots showing the obtained chemical shift offset ($\Delta\omega$) values from paramagnetic guest exchange saturation transfer (paraGEST) experiments with different lanthanide-cyclodextrins (**Ln-CDs**) as the hosts and either **2** (**a**) or **3** (**b**) as the guest, and the assigned colors for each host–guest pair (assigned at Supplementary Tables 3 and 8); **c** CODE-HD color map obtained from different pairs of **Ln-CD** and **3**, which is deleted upon anthranilic acid (**5**, red star) addition to each one of the wells in the plate. The z-spectra obtained from the well containing erbium-cyclodextrin (**Er-CD**, gray diamond) before and after the addition of **5** is shown in the insets; CODE-HD of a sealed three-dimensional (3D) multilayer object. **d** Scanned object scheme, composing of three 3 × 3 multiwell plates, and different **Ln-CD:2** pairs loading patterns (diamonds), **e** complete object and MRI schemes, reflecting the inability to discover the content of each layer and **f** generated CODE-HD maps (rectangles) of each layer.

level, by utilizing different types of modifiable and erasable components. Moreover, it shows its potential to be further extended with additional putative guests in the future.

Finally, to emphasize another advantage of CODE-HD over other approaches, where luminescent colors play a pivotal role, we capitalized on the tomographic capabilities of MRI, which enables the spatial display of maps of different planes of a scanned (sealed and opaque) object. For that purpose, we designed and manufactured a sealed three-dimensional (3D) apparatus that is composed of three layers of 3 × 3 multiwell plates, thereby gaining 27 different compartments, which were loaded with different **Ln-CD:2** (Fig. 5d and Supplementary Table 10) to create three distinct 3 × 3 patterns (Fig. 5d). Assembling and sealing the three plates into a single multilayer object, and activating CODE-HD using an MRI scanner (Fig. 5e), benefited us with the feature of virtual slice selection, allowing to obtain three distinctive multicolored patterns that were encoded by the layers of the scanned opaque object (Fig. 5f). If used for molecular steganography, this ability increases the number of possible permutations and thus authorizes an increased complexity for encoding information, where each layer is translated to a different component of the message. Importantly, the abilities of MRI-based scans to spatially map an object from three different planes (namely, sagittal, axial, and coronal) and virtually section it to longitudinal, lateral, and vertical planes, highlights another unique feature of CODE-HD to encode and decode information in a multitude of modes in the future. These modes will allow the simultaneous loading of single or multiple paraGEST libraries in a given setup, and thus enabling mapping of even more complex arrays.

In summary, we show here the design, development, principles, and implementation of a nonluminescent supramolecular system, CODE-HD, which enables 1D (NMR), 2D, and 3D displays of

artificial colors. By synthesizing a library of paramagnetic cavitands that are based on a lanthanide-cradled α-CD, identifying putative fluorinated guests, and combining host–guest-binding kinetics with paraCEST and $^{19}$F-MRI, the principles by which CODE-HD is operated are established. By inducing PCSs to the chemical shift of dynamically exchanging fluorinated guests in a $^{19}$F-NMR framework, which is controlled by the lanthanide element of the paramagnetic host, spectrally resolved artificial colors are obtained. Following its implementation in $^{19}$F-MRI to allow the spatial display of the artificial colors, the potential use of CODE-HD is demonstrated in several setups. The number of color display permutations resulting from one library of nine pseudo-colors is 387,420,489 ($9^9$), when arranged in a 3 × 3 well plate. However, it can be readily extended by adapting an alternative guest, generating different spectrally resolved pseudo-colors, or by using compartmentalized arrays that generate additional color combinations. While adopting the principles of paraCEST provides CODE-HD with a high spectral resolution and well-resolved colors, its host–guest landscape affords versatility (using a variety of hosts and multiple guests) for color-code switch ability and eraseability. Relying on MRI basics for data acquisition and spatial display of the artificial colors, CODE-HD is exemplified for molecular steganography applications, demonstrating its 3D potential by virtually decoding a multilayered object. While colors that are based on luminescence (fluorescence, chemiluminescence, phosphorescence, etc.) are still the obvious choice for distinguishing between objects, spectrally resolved, light-alternative "colors" can extend the available "multicolor" toolbox for scenarios that are not amenable to the use of light (i.e., opaque objects, media with strong light-scattering or absorbance, photobleaching, limited number of colors, limited spectral resolution, etc.). Thus, we envision that

CODE-HD will be utilized towards multiplexed displays in various fields in basic and applicative sciences.

## Data availability

The data that supports the findings of this study are available upon a reasonable request from the corresponding author.

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

## Acknowledgements

This project has received funding from the European Research Council (ERC) under the European Union's Horizon 2020 research and innovation program (grant agreement No. 677715). The authors thank Dr. Alla Falkovich for her help with the analysis of the MS data.

## Author contributions

E.G. and A.B.-S. designed the study. E.G. synthesized and purified all Ln-CDs. E.G. and L.A. designed and performed all NMR experiments, and analyzed the NMR spectra. E.G. performed all GEST experiments and analyzed the data shown. E.G. and A.B.-S. wrote the manuscript.

## Competing interests

The authors declare no competing interests.
