## [Peer Review File · Nature Communications]

REVIEWERS' COMMENTS

Reviewer #1 (Remarks to the Author):

The authors describe the induction of pseudo-contact shifts in the ^{19}F spectrum of guests that are bound to lanthanide-cradled cyclodextrin hosts. The approach enabled the creation of highly differentiated signals that depend on the lanthanide and the guest species, based on paramagnetic guest exchange saturation transfer (paraGEST). The responses were assigned a pseudo-color and thereby color maps were created that do not depend on optical phenomena such as light absorption or emission. This was used for steganography, including code manipulation and tomographic analysis of objects.

The described implication of paraGEST in ^{19}F NMR and the developed applications are interesting. However, the enabling technology is known in the literature and the advantages of host-guest complexes for designing reversible stimuli-responsive systems have been exemplified in numerous examples. The herein reported expansion is an interesting case study and the linkage to pseudo-color maps is a smart move (albeit such protocols are rather usual in imaging). Every system that is capable of generating signals with a high degree of diversification has potential value for steganography. Without surprise the herein described library is not an exception.

I think this is a nice work, well described and competently performed. However, the demonstrated conceptual advance appears too limited for publication in *Nature Communications*.

Reviewer #2 (Remarks to the Author):

The chemical exchange saturation transfer method is applied to ^{19}F nuclei in guests held within a CD cavity by various paramagnetic ions embedded in the base of the host. A limitation is that the guests need to be outfitted with a primary amine functionality in order to be well-recognized by this technique. Still, a range of guests and lanthanide ions become members of the library. This is a nice extension of a general MRI technique to samples in multi-well plates, even though false-colouring of segments of the range of any variable is common across science. The application to steganography fits well with currently available molecular systems and goes beyond them in terms of the multi-layer capabilities offered. Erasing the data by displacing the fluorinated guest and the tomographic capability are extra positive features which are missing in many current molecular systems. However, the overall weakness of the method is the need for an MRI instrument. But this is not a fault of the method.

CD-DTPA and its lanthanide complexes are known (*J. Am. Chem. Soc.* 1996, 118, 7414). This should be acknowledged in the citations.

Figure 2 line 3; from an aqueous

Figure 4; A relevant reference for molecular generation of graphics displays is *ACS Synth. Biol.* 2020, 9, 1490.

Reviewer #3 (Remarks to the Author):

I quite like the idea of this manuscript, which is to use a supramolecular chemistry approach to access a color-based spatial display, and I recommend that the manuscript be accepted for publication after the following, relatively minor issues are addressed:

1. In the introduction, the authors use the phrase "light-based color." They should clarify the intended meaning of this phrase.
2. In general, the introduction has language that is out of place in a technical scientific paper. The authors refer to an "endless number of innovative supramolecular platforms," as well as the fact that "synthetic supramolecular systems are at the core of advanced fields in science." These and similar statements are hyperbolic and should be modified to more accurately reflect the scientific reality.
3. The supporting information does not include any NMR information for the various cyclodextrin functionalized materials. This omission is curious and should be addressed; while the mass spectra shown certainly match with the masses of the target compounds, the NMR spectra of the compounds before they participate in host-guest inclusion and the associated GEST phenomena should be provided.

Reviewer #4 (Remarks to the Author):

This highly focused communication presents a supramolecular chemistry based technique named CODE-HD for the development of paramagnetic host-guest complexes that through the use of MRI can be used to develop advanced (artificial) colour patterns that can be used to develop codes and patterns that are both readable and erasable. This is a very nice idea and offers the generation of a large number of patterns that can be employed in various research areas as well as in communication of encrypted information and data, etc. I really enjoyed this communication, the idea is clearly outlined and the results seem to support the authors design very nicely. The use of the lanthanides allows the authors to generate the library of the CD-hosts, that they then employed to form the Host-Guest complexes with 4 different guests, three of which are able to demonstrate the necessary shift, from which that colour patterns, using magnetization-transfer ¹⁹F-MRI, results were generated from; the authors using guest 2 to demonstrate this in a very nice way. I think this paper is quite nice, focused as I mentioned, but there are few points I have that the authors need to address before this is published. I would include more description on the synthesis and the characterization of the host and the Ln-complexes in the main document. The Ln-CDs have been characterized by HRMS but no CHN is given, or other in-depth analysis. A statement that the HRMS for these LN-CD matched the calculated isotopic patterns for the relevant complexes might be good to include.

A comment on the stability of the Ln-complexes for the various Ln-CDs should be included. If binding constants are known these should be included too. LN-CDs are well known and some of that work dates back till the 1990s. Some ref. to that should be included, as these types of complexes are normally not employed in the clinic due to low stability.

Were Ce-CD and Sm-CD, that showed relatively small $\Delta\omega$ values, tested in the experiences carried out using the MRI scanner (as was proposed they could be employed for)?

The ref. in SI are not given in full.

Point-by-point response to the Reviewers' comments

Re: Revised Manuscript NCOMMS-20-39754

Title: "Supramolecular Palette: Versatile Non-luminescent Colors based on Guest Exchange Dynamics in Paramagnetic Cavitands"

Reviewer #1

RI.1. I think this is a nice work, well described and competently performed. However, the demonstrated conceptual advance appears too limited for publication in Nature Communications.

The Reviewer's comment suggests that the conceptual advance of our manuscript might not have been emphasized enough in the first submitted version of our manuscript. We would like to point out that, although artificial colors for MRI-based studies have been proposed in the past, including with CEST and paraCEST, the combination of supramolecular chemistry, paraCEST and ^{19}F -MRI was not proposed before. This combination enables color-display features that are not yet available either for MRI-based colors, *e.g.*, (i) color palette extendibility (additional hosts and guests), (ii) transformability (replacing one guest with another) and (iii) color-display erasability, or luminescent-colors, *e.g.*, (i) tolerability for opaque objects and (ii) the ability to virtually slice multi-layered objects for the simultaneous display of complexed sealed arrays. Therefore, we truly believe that the demonstrated approach is robust and conceptually novel, yet general.

Reviewer #2

R2.1. A limitation is that the guests need to be outfitted with a primary amine functionality in order to be well-recognized by this technique. Still, a range of guests and lanthanide ions become members of the library.

We thank the Reviewer for highlighting this point. To elaborate on this issue, we examined an additional guest, *i.e.*, 4-fluorobenzoic acid, in order to emphasize the potential of CODE-HD to be further extended with additional guests. This additional demonstration, and the fact that both the primary-amine and the carboxylate functionalities of the fluorinated guests were found to be applicable for CODE-HD, highlight the necessity of functional groups with Ln^{3+} -coordination capabilities for obtaining the reported paraGEST phenomena. This interpretation implies that additional guests with Ln^{3+} -coordination capabilities, such as phosphate, sulfonate, thiolates and even pyridine, should be considered in the future in order to extend the putative library shown in our manuscript.

The following discussion was added to the text: "Importantly, an analog of guest **1**, which is substituted with a carboxylic acid instead of a primary amine group, also yielded a pronounced paraGEST effect (Figure S41a), while other analogs, substituted with hydroxyl or amino-methyl functional groups, did not generate any noted effect (Figure S41b-d). These observations reflect the cruciality of a functional group with Ln^{3+} -coordination capabilities (primary amine or a carboxylic acid) for a successful magnetization transfer effect (Figure 2 and S41)."

R2.2. The overall weakness of the method is the need for an MRI instrument. But this is not a fault of the method.

We agree with the Reviewer that the need for an MRI is an intrinsic limitation of the method. However, as emphasized in the conclusions, CODE-HD can be also applied with a conventional NMR setup, which is available at any research institute. Thus, it can be widely implemented by many researchers, but with the limitation of 1D encoding feature (NMR) and without the 2D display capabilities, which, indeed, require an MRI instrument, as mentioned by the Reviewer. "In summary, we show here the design, development, principles and implementation of a non-luminescent supramolecular system, CODE-HD, which enables 1D (NMR), 2D, and 3D displays of artificial colors."

R2.3. CD-DTPA and its lanthanide complexes are known (J. Am. Chem. Soc. 1996, 118, 7414). This should be acknowledged in the citations.

We thank the Reviewer for highlighting the previously reported CD-DTPA lanthanide complexes, which should have been acknowledged in our first submission. We have now added two new references (i.e., references 44 and 45) to highlight the originally proposed Ln-CDs.

R2.4. Figure 2 line 3; from an aqueous.

We thank the Reviewer for this comment. This has been corrected.

R2.5. Figure 4; A relevant reference for molecular generation of graphics displays is ACS Synth. Biol. 2020, 9, 1490.

We thank the Reviewer for highlighting this very important example of a molecular generation of graphic displays. The relevant reference was added as the new reference 55.

Reviewer #3

R3.1. In the introduction, the authors use the phrase “light-based color.” They should clarify the intended meaning of this phrase.

We thank the Reviewer for highlighting this point. This phrase was replaced with “luminescent color” throughout the text.

R3.2. . In general, the introduction has language that is out of place in a technical scientific paper. The authors refer to an “endless number of innovative supramolecular platforms,” as well as the fact that “synthetic supramolecular systems are at the core of advanced fields in science.” These and similar statements are hyperbolic and should be modified to more accurately reflect the scientific reality.

We thank the Reviewer for pointing this out. These phrases were modified to more accurately reflect the scientific reality, as requested.

- (a) “...establishing synthetic supramolecular systems as accessible tools for both fundamental research and advanced applications”
- (b) “...created the opportunity to design a large number of innovative supramolecular platforms”

R3.3. The supporting information does not include any NMR information for the various cyclodextrin functionalized materials. This omission is curious and should be addressed; while the mass spectra shown certainly match with the masses of the target compounds, the NMR spectra of the compounds before they participate in host-guest inclusion and the associated GEST phenomena should be provided.

We thank the Reviewer for highlighting this very important point, which was not clearly shown in the first version of the manuscript. As requested by the Reviewer, we performed extensive 1D (¹H-NMR, ¹³C-NMR) and 2D (¹H-¹H COSY, ¹H-¹³C HSQC and ¹H-¹H ROESY) NMR experiments to generate a new set of NMR spectra for the studied cavitand (CD-DTPA) that participate in the host-guest inclusion and the associated GEST (new Figures S2-S19).

The following sentence was added to the text: “The purity of the obtained **CD-DTPA** product was evaluated by an analytical HPLC (Figure S1), followed by its full characterization using a set of 1D (¹H-, ¹³C-) and 2D (¹H-¹H COSY, ¹H-¹³C HSQC and ¹H-¹H ROESY) NMR experiments (Figures S4-S19 and Appendix A)...”

A relevant explanation was added to the SI file as Appendix A: “While native CDs exhibit high symmetry, which

allows a relatively easy characterization by NMR, partial modifications on their glucose units can lead to the splitting of the spin systems due to symmetry breaks. This results in a crowded spectrum with an overly-split, overlapped, broad aliphatic area, as evident only from Diamino-CD's NMR spectra (Figures S2 and S3).

The additional modification with DTPA had caused splitting to three spin-systems in **CD-DTPA** (Figures S4-S8), which can be mainly evident by the location of three representing peaks for the anomeric H-1 protons (i.e., H-1, H-1' and H-1'', Figure S4, 4.99-5.05 ppm), indicating three types of CD's sugar units. To complete the characterization, advanced 2D-NMR experiments, such as COSY, HSQC and ROESY (Figures S9-S18), were performed. 2D COSY NMR (Figures S9-S12) had allowed us to first identify the CD-DTPA's inner protons (H-1 to H-6) and assign their "ranges". Following that, 2D HSQC NMR (Figures S13-S16) had enabled to distinguish between most of CD-DTPA's inner protons/carbons (H/C-1 to H/C-5) to the DTPA-bridge's protons/carbons (H/C-7 to H/C-11) and H-6, based only on the correlations' phases (i.e., green and blue, representing primary and secondary carbons respectively). Moreover, this experiment indicates the existence of inequivalent protons on several carbons (C-6, C-8 and C-9, indicated by "a" and "b"). The two H-6 protons undergo dramatic changes in their chemical shifts due to the DTPA modification. 2D ROESY NMR (Figure S17 and S18) had enabled the identification of an H-4 peak underlying H-2's multiplet, due to a "through-space" interaction between H-4 and H-1.

Considering the intricate NMR spectra and broadside chemical shifts (summarized at Figure S19), characterization was mainly based on MS analysis (Figures S21-S42)."

Reviewer #4

R4.1 I would include more description on the synthesis and the characterization of the host and the Ln-complexes in the main document. The Ln-CDs have been characterized by HRMS but no CHN is given, or other in-depth analysis. A statement that the HRMS for these LN-CD matched the calculated isotopic patterns for the relevant complexes might be good to include.

We thank the Reviewer for this suggestion. As recommended by the Reviewer, we have added the description of the synthesis to the main document: "Following 16 hours of stirring in anhydrous dimethyl sulfoxide and triethylamine, the resultant white powder was collected after centrifugation and purified using preparative reversed-phase high-pressure liquid chromatography (HPLC). The purity of the obtained CD-DTPA product was evaluated by an analytical HPLC (Figure S1), followed by its full characterization using a set of 1D (¹H-NMR, ¹³C-NMR) and 2D (¹H-¹H COSY, ¹H-¹³C HSQC and ¹H-¹H ROESY) NMR experiments (Figures S4-S19 and Appendix A) and high-resolution mass spectrometry (Figures S20 and S21). Then, based on the procedure applied to obtain lanthanide-cradled β-cyclodextrins,^{44,45} pure CD-DTPA was refluxed in water..."

In addition, as requested by the Reviewer, we have added an in-depth analysis of the products, including:

(i) Elemental CHN analysis (Table S1 For Reviewer).

The content of C, H and N atoms in the sample (36.67%, 4.11% and 4.11% respectively) was found to be smaller than theoretically calculated for C₅₀H₈₁N₅O₃₆ (45.21%, 6.15% and 5.27% respectively). A ¹⁹F-NMR spectrum was acquired and revealed a 1:1 ratio between CD-DTPA and TFA (from the HPLC eluent based on the integration of the representing peak for TFA, -76 ppm, and a previously calibrated Sodium fluoride capillary). Fitting the data into a JavaScript Percentage Elemental Results Calculator (JASPER), while limiting the number of TFA molecules to one, resulted a chemical formulation of C₅₀H₈₁N₅O₃₆·1TFA·9.2H₂O (38.84%, 6.29% and 4.36% respectively).

To the best of our understanding, elemental analysis of macrocycles might not be accurate, due to the presence of bound water molecules, and in particular for CDs, which includes water inside their cavity, and are sensitive to moisture. Thus, we examined the C/N ratio in the measured sample (8.92) and compared it to the calculated formula C₅₀H₈₁N₅O₃₆·1TFA·9.2H₂O (8.91) to characterize the compound. For this reason, the new results were not reported in the text and are added for the Reviewer revision. Nevertheless, it should be emphasized here that we have performed an extensive NMR studies (see point *ii* next) to characterize our synthetic host in full.

(ii) Extensive NMR experiments, including 1D (¹H-NMR, ¹³C-NMR) and 2D (¹H-¹H COSY, ¹H-¹³C HSQC and ¹H-¹H ROESY) NMR experiments to further characterize the newly proposed compounds (New Figures S4-S19 and Appendix A). The following sentence was added to the text: "The purity

of the obtained **CD-DTPA** product was evaluated by an analytical HPLC (Figure S1), followed by its full characterization using a set of 1D (^1H -, ^{13}C -) and 2D (^1H - ^1H COSY, ^1H - ^{13}C HSQC and ^1H - ^1H ROESY) NMR experiments (Figures S4-S19 and Appendix A)...”

A relevant explanation was added to the SI file as Appendix A: “While native CDs exhibit high symmetry, which allows a relatively easy characterization by NMR, partial modifications on their glucose units can lead to the splitting of the spin systems due to symmetry breaks. This results in a crowded spectrum with an overly-split, overlapped, broad aliphatic area, as evident only from Diamino-CD’s NMR spectra (Figures S2 and S3).

The additional modification with DTPA had caused splitting to three spin-systems in **CD-DTPA** (Figures S4-S8), which can be mainly evident by the location of three representing peaks for the anomeric H-1 protons (i.e., H-1, H-1’ and H-1’’, Figure S4, 4.99-5.05 ppm), indicating three types of CD’s sugar units. To complete the characterization, advanced 2D-NMR experiments, such as COSY, HSQC and ROESY (Figures S9-S18), were performed. 2D COSY NMR (Figures S9-S12) had allowed us to first identify the CD-DTPA’s inner protons (H-1 to H-6) and assign their “ranges”. Following that, 2D HSQC NMR (Figures S13-S16) had enabled to distinguish between most of CD-DTPA’s inner protons/carbons (H/C-1 to H/C-5) to the DTPA-bridge’s protons/carbons (H/C-7 to H/C-11) and H-6, based only on the correlations’ phases (i.e., green and blue, representing primary and secondary carbons respectively). Moreover, this experiment indicates the existence of inequivalent protons on several carbons (C-6, C-8 and C-9, indicated by “a” and “b”). The two H-6 protons undergo dramatic changes in their chemical shifts due to the DTPA modification. 2D ROESY NMR (Figure S17 and S18) had enabled the identification of an H-4 peak underlying H-2’s multiplet, due to a “through-space” interaction between H-4 and H-1. Considering the intricate NMR spectra and broadside chemical shifts (summarized at Figure S19), characterization was mainly based on MS analysis (Figures S21-S42).”

- (iii) As requested by the Reviewer, the HR-MS data is now accompanied by calculated isotopic patterns, which show good correlation with the experimental spectra obtained for the Ln-CDs (New Figures S21, S22-S40).

R4.2. A comment on the stability of the Ln-complexes for the various Ln-CDs should be included. If binding constants are known these should be included too.

As recommended by the Reviewer, we have added a comment on the stability of the obtained Ln-complexes: “Note here that the obtained octadentate aminopolycarboxylate CD-DTPA should have a strong binding affinity towards lanthanides in the resultant Ln-CDs, as those obtained for the clinically-used Gd-DTPA-BMA contrast agent⁴⁶. Nevertheless, as the binding affinities may slightly deviate for different Ln^{3+} , as reported for different Ln-DTPA-BMAs^{47, 48}, further stability studies and safety profiles should be obtained prior to the use of Ln-CDs in any biological application in the future.” New relevant references (46-48) were added as well.

R4.3. Were Ce-CD and Sm-CD, that showed relatively small $\Delta\omega$ values, tested in the experiences carried out using the MRI scanner (as was proposed they could be employed for)?

We thank the Reviewer for this comment. To elaborate on the potential use of both Ce-CD and Sm-CD in CODE-HD and their function as additional artificial colors in future applications, both hosts were examined using a 15.2 T MRI scanner, the results of which now appear as new Figure S45. An additional comment was added to the text as well: “Note here that future applications including **Ce-CD** and **Sm-CD**, which induced relatively small $\Delta\omega$ values in **2** (Figure S43), can add additional colors to CODE-HD, enabling even more color combinations (although limited for strong-field MRI scanners, as the one used here, Figure S45). By this, CODE-HD is able to offer, in principle, eleven factorial ($11! = 39,916,800$) different combinations in dedicated setups.”

R4.4. The ref. in SI are not given in full.

This was corrected.

New Figures

New Figure S1. CD-DTPA absorbance at 220 nm. Measured at analytical HPLC.

New Figure S2. $^1\text{H-NMR}$ spectrum (500.08 MHz) for Diamino-CD.

New Figure S3. $^{13}\text{C}\{^1\text{H}\}$ NMR spectrum (125.75 MHz) for Diamino-CD.

New Figure S4. ¹H-NMR spectrum (500.008 MHz) for CD-DTPA.

New Figure S5. Region of ¹H-NMR spectrum (Figure S4) showing peaks for H-2 to H-11 of CD-DTPA.

New Figure S6. $^{13}\text{C}\{^1\text{H}\}$ NMR spectrum (125.75 MHz) for CD-DTPA. * Peaks representing TFA contaminations.

New Figure S7. Region of $^{13}\text{C}\{^1\text{H}\}$ NMR spectrum (Figure S6) showing peaks of C-1 to C-11.

New Figure S8. Part of $^{13}\text{C}\{^1\text{H}\}$ -NMR spectrum (Figure S6) showing peaks for three types of C-2, C-3 and C-5.

New Figure S9. ^1H - ^1H COSY spectrum (500.008 MHz) for CD-DTPA.

New Figure S10. Region of ^1H - ^1H COSY spectrum (Figure S9) showing the interactions between H-1, H-1', H-1'' (F2) and H-2, H-2', H-2'' (F1).

New Figure S11. Part of ^1H - ^1H COSY spectrum (Figure S9) showing interactions between CD-DTPA's ring protons (H-2 to H-5).

New Figure S12. Part of ^1H - ^1H COSY spectrum (Figure S9) showing interactions between CD-DTPA's bridge protons H-8_{a/b} and H-9_{a/b}.

New Figure S13. ^1H - ^{13}C HSQC spectrum (500.008 MHz, 125.75 MHz) for CD-DTPA.

New Figure S14. Region of ^1H - ^{13}C HSQC spectrum (Figure S13) showing H/C-1 correlations.

New Figure S15. Region of ^1H - ^{13}C HSQC spectrum (Figure S13) showing CH_2 correlations (H/C-6 to H/C-11).

New Figure S16. Part of ^1H - ^{13}C HSQC spectrum (Figure S13) magnification showing CH correlations (H/C-2 to H/C-5).

New Figure S17. ^1H - ^1H ROESY spectrum (500.008 MHz) for CD-DTPA.

New Figure S18. Region of ^1H - ^1H ROESY spectrum (Figure S17) showing H-1 (F1) and H-4 (F2) correlations.

b

Proton Number	Chemical shift [ppm]
H-1	4.94
H-1', H-1''	4.99, 4.98
H-2, H-2', H-2''	3.61 - 3.52
H-3, H-3', H-3''	3.95 - 3.85
H-4	3.61 - 3.52
H-4', H-4''	3.40 - 3.32
H-5	3.85 - 3.80
H-5', H-5''	3.80 - 3.72
H-6 _a	3.32 - 3.22
H-6 _b	4.02 - 3.99
H-6' _a , H-6'' _a	3.65 - 3.59
H-6' _b , H-6'' _b	3.72 - 3.65
H-7	4.14 - 4.04
H-8 _{a+b}	3.22 - 3.09
H-9 _{a+b}	3.47 - 3.36
H-10	3.95 - 3.85
H-11	3.61 - 3.52

c

Carbon Number	Chemical shift [ppm]
C-1	101.48
C-1', C-1''	101.60, 101.51
C-2	71.53
C-2', C-2''	71.75
C-3	72.81
C-3', C-3''	73.44, 73.39
C-4	83.90
C-4', C-4''	82.05, 81.01
C-5	72.13
C-5', H-5''	71.62
C-6	40.82
C-6', C-6''	60.68, 56.78
C-7	54.57
C-8, C-9	50.60
C-10	60.12
C-11	54.58
C-12 - C-14	170.10, 170.65, 173.05

New Figure S19. CD-DTPA NMR assignment. (a) CD-DTPA structure and atoms indications; (b) ¹H-NMR assignment; (c) ¹³C-NMR assignment.

New Figure S20. Isotopic Patterns for CD-DTPA. (a) Simulated mass distribution; (b) Experimental mass distribution.

New Figure S21. MS spectrum and isotopic patterns for Dy-CD. (a) Full range MS chromatogram; (b) Simulated mass distribution; (c) Experimental mass distribution.

New Figure S22. MS spectrum and isotopic patterns for Tb-CD. (a) Full range MS chromatogram; (b) Simulated mass distribution; (c) Experimental mass distribution.

New Figure S23. MS spectrum and isotopic patterns for Yb-CD. (a) Zoom for $[M]^+$ peak range.; (b) Simulated mass distribution; (c) Experimental mass distribution.

New Figure S24. Isotopic Patterns for Ho-CD. (a) Simulated mass distribution; (b) Experimental mass distribution.

New Figure S25. Isotopic Patterns for Eu-CD. (a) Simulated mass distribution; (b) Experimental mass distribution.

New Figure S26. MS spectrum and isotopic patterns for La-CD. (a) Full range MS chromatogram; (b) Simulated mass distribution; (c) Experimental mass distribution.

New Figure S27. MS spectrum and isotopic patterns for Er-CD. (a) Full range MS chromatogram (D₂O); (b) Simulated mass distribution (H₂O); (c) Simulated mass distribution (D₂O); (d) Experimental mass distribution.

New Figure S28. MS spectrum and isotopic patterns for Tm-CD. (a) Full range MS chromatogram (D₂O); (b) Simulated mass distribution (D₂O); (c) Simulated mass distribution (H₂O); (d) Experimental mass distribution.

New Figure S29. Isotopic Patterns for Pr-CD. (a) Simulated mass distribution; (b) Experimental mass distribution.

Figure S30. Isotopic patterns for Sm-CD. (a) Simulated mass distribution (D_2O); (b) Simulated mass distribution (H_2O); (d) Experimental mass distribution.

New Figure S31. MS spectrum and isotopic patterns for Ce-CD. (a) Zoom for $[M]^+$ peak range (D_2O); (b) Simulated mass distribution (D_2O); (c) Simulated mass distribution (H_2O); (d) Experimental mass distribution.

New Figure S32. Isotopic patterns for Nd-CD. (a) Simulated mass distribution (D₂O); (b) Simulated mass distribution (H₂O); (d) Experimental mass distribution.

Updated Figure S33. z-spectra of paraGEST screening experiments for variable guests: (a) 4-Fluorobenzoic acid, (b) 4-Fluorobenzylalcohol, (c) 4-Fluoro-N-methyl-Benzenemethanamine and (d) 4-Fluoro-phenylalanine.

New Figure S34. paraGEST MRI experiments for Ce-CD and Sm-CD. (a) ^1H -MRI map of four 4 cm tubes filled with Ce-CD:2 in 1:100 ratio (ROI 1), Sm-CD:2 in 1:5 ratio (ROI 2), guest 2 without host (ROI 3) and water (ROI 4); (b) ^{19}F -MRI map depicting only tubes with guest 2; (c) ^{19}F -paraGEST MRI contrast overlaid on ^1H -map revealing the tubes filled with Ce-CD (green) and Sm-CD (orange); z-spectra measured for (d) ROI 1, (e) ROI 2 and (f) ROI 3.

Data for Reviewers

Formula	Element Content (% by weight)			C/N Ratio
	C	H	N	
$C_{50}H_{81}N_5O_{36}$ - Theoretical	45.21	6.15	5.27	-
$C_{50}H_{81}N_5O_{36}$ - Experimental	36.67 ± 0.07	4.11 ± 0.60	4.11 ± 0.06	8.92
$C_{50}H_{81}N_5O_{36} \cdot 1$ TFA	43.31	5.73	4.86	8.91
$C_{50}H_{81}N_5O_{36} \cdot 1$ TFA $\cdot 9.2$ H ₂ O	38.84	6.29	4.36	8.91

Table 1 for Reviewers. Elemental Analysis for CD-DTPA. Theoretical and experimental element content for Carbon (C), Hydrogen (H) and Nitrogen (N) atoms in CD-DTPA and formulae calculated for CD-DTPA with TFA and H₂O impurities.

REVIEWERS' COMMENTS

Reviewer #3 (Remarks to the Author):

The authors have done an outstanding job responding to my comments on the previous version of their manuscript. I am pleased at this point to recommend that the article be accepted for publication.